# Exploring the Role of the Gut Microbiota in Modulating Colorectal Cancer Immunity

**DOI:** 10.3390/cells13171437

**Published:** 2024-08-27

**Authors:** Nikolay K. Shakhpazyan, Liudmila M. Mikhaleva, Arkady L. Bedzhanyan, Zarina V. Gioeva, Alexander I. Mikhalev, Konstantin Y. Midiber, Valentina V. Pechnikova, Andrey E. Biryukov

**Affiliations:** 1Avtsyn Research Institute of Human Morphology, Petrovsky National Research Center of Surgery, 119435 Moscow, Russia; mikhalevalm@yandex.ru (L.M.M.); gioeva_z@mail.ru (Z.V.G.); midiberkonst@gmail.com (K.Y.M.); valiagtx@yandex.ru (V.V.P.); bervost@rambler.ru (A.E.B.); 2Department of Abdominal Surgery and Oncology II (Coloproctology and Uro-Gynecology), Petrovsky National Research Center of Surgery, 119435 Moscow, Russia; arkady.bedzhanyan@gmail.com; 3Department of Hospital Surgery No. 2, Pirogov Russian National Research Medical University, 117997 Moscow, Russia; rsmu1985@gmail.com; 4Institute of Medicine, Peoples’ Friendship University of Russia named after Patrice Lumumba, 6 Miklukho-Maklaya St., 117198 Moscow, Russia

**Keywords:** gut microbiota, colorectal cancer, immune modulation, immunotherapy, microbial diversity

## Abstract

The gut microbiota plays an essential role in maintaining immune homeostasis and influencing the immune landscape within the tumor microenvironment. This review aims to elucidate the interactions between gut microbiota and tumor immune dynamics, with a focus on colorectal cancer (CRC). The review spans foundational concepts of immuno-microbial interplay, factors influencing microbiome composition, and evidence linking gut microbiota to cancer immunotherapy outcomes. Gut microbiota modulates anti-cancer immunity through several mechanisms, including enhancement of immune surveillance and modulation of inflammatory responses. Specific microbial species and their metabolic byproducts can significantly influence the efficacy of cancer immunotherapies. Furthermore, microbial diversity within the gut microbiota correlates with clinical outcomes in CRC, suggesting potential as a valuable biomarker for predicting response to immunotherapy. Conclusions: Understanding the relationship between gut microbiota and tumor immune responses offers potential for novel therapeutic strategies and biomarker development. The gut microbiota not only influences the natural history and treatment response of CRC but also serves as a critical modulator of immune homeostasis and anti-cancer activity. Further exploration into the microbiome’s role could enhance the effectiveness of existing treatments and guide the development of new therapeutic modalities.

## 1. Introduction

In recent years, the interplay between the human microbiome and cancer immunity has garnered significant attention within the scientific community. The human microbiome, particularly the gut microbiota, is composed of trillions of microorganisms that play a pivotal role in maintaining immune homeostasis and modulating immune responses [1]. The complex interactions between the gut microbiota and the host immune system begin early in life and continue to influence health outcomes throughout adulthood.

The microbiome’s role in cancer immunity is multifaceted, involving direct and indirect mechanisms that can either promote or inhibit tumor development and progression. These mechanisms include modulation of inflammatory responses, enhancement of immune surveillance, and regulation of systemic immune functions. Emerging evidence suggests that specific microbial species and their metabolic byproducts can significantly influence the efficacy of cancer immunotherapies, such as immune checkpoint inhibitors [2,3].

Immunotherapy has revolutionized the treatment landscape for various cancers, including melanoma, non-small cell lung cancer, and renal cell carcinoma [4,5,6]. However, the clinical response to immunotherapy is highly variable, with some patients experiencing remarkable and durable responses while others derive no benefit. This variability underscores the need for novel biomarkers and predictors of response to identify patients who are most likely to benefit from these therapies [3,7].

Recent studies have highlighted the potential of the gut microbiome as a key determinant of immunotherapy response. The composition and functional capacity of the gut microbiota appear to correlate with clinical outcomes in patients receiving immune checkpoint inhibitors. Specific microbial signatures have been associated with enhanced therapeutic efficacy, suggesting that the gut microbiota may serve as a valuable biomarker for predicting response to immunotherapy.

Given the critical role of the microbiome in shaping immune responses and its potential to modulate the effectiveness of cancer treatments, there is a growing interest in understanding the underlying mechanisms that govern microbiome–immune interactions. This review aims to provide a comprehensive overview of the current understanding of the microbiome’s role in anti-cancer immunity, with a particular focus on the implications for immunotherapy. We will explore the foundational concepts of immuno-microbial interplay, the factors influencing microbiome composition and function, and the emerging evidence linking the gut microbiota to cancer immunotherapy outcomes.

## 2. The Limits of Human Tolerance to Microbiota

It is intriguing to evaluate where the immune system draws the line concerning microbial tolerance. While there are several extreme states of gut microbiota, they do not elicit pronounced hyperergic reactions.

Microbial DNA in the bloodstream is one example. Research has detected microbial DNA fragments in the blood of healthy individuals [8,9]. Thought to originate from the gut microbiota, this DNA can enter the bloodstream through various mechanisms, such as microscopic breaches in the gut barrier (sometimes described as “leaky gut”) or uptake by immune cells interacting with gut microbes. The presence of microbial DNA in the bloodstream does not typically trigger an immune response, as these fragments are recognized as non-threatening under normal conditions. This challenges the traditional view of blood as a sterile environment and suggests that low-level bacterial translocation may be a normal aspect of human physiology, potentially playing a role in modulating the immune system and maintaining homeostasis [10].

Another example—microbial metabolites in circulation (SCFAs and Others). These metabolites can traverse the gut barrier, enter the bloodstream, and exert systemic effects, including on the immune system and metabolism [1]. SCFAs, for instance, have been shown to enhance the regulatory functions of T cells and modulate inflammation, demonstrating how substances produced by the gut microbiota can influence systemic health [11].

Lipopolysaccharides (LPSs) are known as a powerful stimulator of innate immune response. LPSs, components of the outer membrane of gram-negative bacteria, can be found at low levels in the bloodstream of healthy individuals. Normally, these low levels of LPSs do not cause harm and may even prime the immune system for a more effective response to bacterial infections [12]. However, elevated levels of LPSs are linked with systemic inflammation and can lead to sepsis if unchecked [13].

Commensal intracellular bacteria: The presence of commensal intracellular bacteria within the gut epithelium is less established compared to pathogenic intracellular bacteria. Nevertheless, some bacteria, typically considered commensal, have demonstrated the ability to enter epithelial cells under specific conditions without causing disease or triggering a strong inflammatory response [14]. These interactions may play roles in educating the immune system and maintaining homeostasis. For example, transient entry into epithelial cells could affect epithelial cell function and local immune responses, potentially contributing to tolerance mechanisms against the gut microbiota [15,16].

This exploration of the limits of immune tolerance to the microbiota reveals that the human body can maintain a delicate balance with its microbial inhabitants under even extreme conditions. The presence of microbial DNA in the bloodstream, circulating microbial metabolites, and even commensal intracellular bacteria highlight a sophisticated level of immune regulation that allows for both tolerance and vigilance. These interactions are pivotal for maintaining homeostasis and educating the immune system, suggesting that our understanding of sterility and immune reactivity may need to be redefined.

Since later in the review we will discuss tumor cells that initially originated from normal cells of the body, the ability of microbes to be tolerated despite being completely foreign initially is demonstrated. In conclusion, we can make a brief lyrical insert: “What was native has become hostile, what was alien has become a friend”.

## 3. Factors Influencing the Impact of the Gut Microbiome on Immunity

The complex relationship between the human microbiome and immune modulation is valuable, particularly in understanding how the gut microbiota influences immune responses. The roles of the microbiome extend from early immune development to the establishment of microbial tolerance in adulthood. This exploration is crucial given the emerging evidence linking microbial diversity and immunotherapy efficacy in diseases like colorectal cancer. This section details the roles of key bacterial species and the modifying effects of diet and antibiotics, aiming to clarify the mechanisms through which the gut microbiome impacts health. These discussions highlight the potential for microbiome modulation to improve treatment outcomes in conditions mediated by immune responses.

Microbial diversity: High microbial diversity is correlated with a robust immune system. Diverse microbiota enhances the production of short-chain fatty acids (SCFAs) such as butyrate, crucial for maintaining gut barrier integrity and modulating immune responses. Butyrate specifically promotes the differentiation and function of regulatory T cells (Tregs) in patients with autoimmune conditions through mTOR-mediated pathways, vital for maintaining immune balance and suppressing inflammatory responses, particularly by restoring Treg subpopulations [17].

There are notable bacterial species including *Faecalibacterium prausnitzii*, *Bacteroides fragilis*, Bifidobacterium species, Lactobacillus species, *Akkermansia muciniphila*, and *Prevotella copri* that, according to recent data, significantly affect host immunity.

*Faecalibacterium prausnitzii*: This species alleviates inflammatory arthritis and regulates the production of SCFAs and cytokines such as IL-17 in an experimental mouse model of rheumatoid arthritis [18]. It contributes to the modulation of gut inflammation and the maintenance of immune homeostasis through its metabolic activities. Research on humanized mice demonstrated that F. prausnitzii aids in the induction of DP8α Tregs, which protect against intestinal inflammation [19]. Further studies found that it prevents physiological damage in a chronic low-grade inflammation murine model by reducing gut permeability and restoring apical junction proteins, essential for maintaining gut barrier integrity [20].

*Bacteroides fragilis*: Known for its immune modulation and anti-inflammatory effects, it produces polysaccharide A (PSA), which stimulates the immune system, promoting the production of anti-inflammatory cytokines such as IL-10 and enhancing Treg activity [21,22].

Bifidobacterium species: Enhance both innate and adaptive immune responses by modulating gut microbiota composition, stimulating dendritic cell maturation, and promoting the production of cytokines that enhance immune function. These species improve gut health by promoting the production of IgA and enhancing the gut barrier, which helps protect against infections [23,24].

Lactobacillus species: Strengthen the gut barrier and modulate immune responses by producing lactic acid and other antimicrobial substances that inhibit pathogenic bacteria, stimulate mucus production, and enhance the function of Tregs and other immune cells, particularly by inhibiting macrophage pyroptosis [25,26].

*Akkermansia muciniphila*: Maintains mucosal integrity and modulates immune responses by promoting the production of mucins, which protect the gut lining, and stimulating immune cells to produce anti-inflammatory cytokines [27,28,29,30,31].

Role of *Prevotella copri*: The presence of *Prevotella copri* in the gut modifies the beneficial effects of fiber on C-reactive protein levels. Individuals without substantial *P. copri* carriage experienced more significant C-reactive protein reductions with fiber intake, whereas those with *P. copri* did not show these benefits, suggesting that *P. copri* might influence the inflammatory response to dietary fiber [32].

Dietary influences: Diets rich in fiber and polyphenols support a healthy microbiome, whereas high-fat and high-sugar diets can disrupt the microbial balance. In a study involving 307 generally healthy men, gut microbiomes were examined using shotgun metagenomic and metatranscriptomic sequencing. The study linked long-term and recent dietary fiber intake to plasma levels of C-reactive protein (CRP), highlighting the complex interactions between diet and gut microbiota in modulating systemic inflammation. This suggests that dietary interventions, particularly those increasing fiber intake, could mitigate inflammation through microbiota-mediated pathways [32,33].

Antibiotics and medications: Antibiotics reduce microbial diversity, which can impair SCFA production and promote the growth of pathogenic bacteria, triggering inflammatory responses. Experimental studies on mice have shown that antibiotic treatments, such as with vancomycin and polymixin B sulfate, alter the gut microbiota composition and affect local metabolomes and immune responses. These treatments have been linked to the upregulation of both pro-inflammatory (IFN-γ, TNF-α, IL-1β, IL-6) and anti-inflammatory (IL-4) cytokines, as well as effector cytokines of the Th17 cell subset (IL-17 and IL-23), suggesting a direct link between microbial changes and immune dysregulation [34]. In models of experimental colitis, antibiotic-associated dysbiosis impaired the gut microbiota’s ability to control intestinal inflammation, indicating that disruption of the microbiome by antibiotics can exacerbate inflammatory conditions [35].

Host genetics: Genetic factors of the host influence the composition and function of the gut microbiome. A study estimated the SNP-based heritability of various microbiota, finding significant heritability estimates for families like Desulfovibrionaceae and Odoribacter, suggesting that host genetics play a role in determining the composition of the gut microbiome [36,37,38]. The MiBioGen consortium study, involving over 18,000 participants, has highlighted the influence of common genetic factors like the lactase gene (LCT) and the fucosyl transferase gene (FUT2) on the gut microbiome. These genes were found to affect the abundance of key bacterial taxa, illustrating the direct influence of human genetics on microbiome composition, which in turn can affect host metabolism, nutrition, and immunity [39].

In conclusion, the interplay among diverse bacterial species, dietary components, medications, and genetic factors critically shapes the gut microbiome’s influence on host immunity. These findings underscore the microbiome’s essential role in maintaining immune equilibrium. A summary of the chapter is shown in Table 1.

## 4. Molecular Mechanisms Encoded by Host and Microbial Genes in Gut Microbiome–Immune System Interactions

Genetics plays a critical role in the complex interplay between the gut microbiome and the host’s immune system. This section highlights key genetic factors that influence these interactions. Host genes, whose products are most prominently involved in interactions with the microbiota, include regulators of innate and adaptive immune mechanisms, adhesion molecules, and factors maintaining epithelial barrier integrity. The most notable ones are listed below.

Pattern recognition receptors (PRRs): Toll-like receptors (TLRs) such as TLR1, TLR2, TLR4, TLR5, and TLR9 recognize bacterial lipoproteins, lipopolysaccharides, flagellin, and bacterial DNA, respectively, contributing to the immune response or tolerance. Interestingly, even closely related bacterial species can activate different types of Toll-like receptors. For instance, Burkholderia mallei primarily activates TLR4, while *Burkholderia pseudomallei* predominantly activates TLR2. Different strains of *Francisella tularensis* interact with TLRs differently, with the highly virulent Schu S4 strain not activating TLR2 or TLR4, whereas less virulent strains do [40,41]. NOD-like receptors (NLRs), including NOD1 and NOD2, detect peptidoglycan motifs found in the cell walls of Gram-positive and Gram-negative bacteria, playing a significant role in immune detection and response. Activation of NOD1 and NOD2 leads to the production of pro-inflammatory cytokines, contributing to the host’s immune defense. It is known that species such as Shigella flexneri, *Escherichia coli*, *Helicobacter pylori*, *Pseudomonas aeruginosa*, *Campylobacter jejuni*, and *Clostridium difficile* interact with NOD1 through peptidoglycan fragments containing gamma-glutamyl diaminopimelic acid (iE-DAP). *Listeria monocytogenes*, *Salmonella enterica Typhimurium*, *Mycobacterium tuberculosis*, and *Mycobacterium bovis* BCG interact with NOD2 through muramyl dipeptide (MDP) present in their peptidoglycan [42].

Tight junction proteins: Claudins, such as claudin-2 and claudin-15, regulate paracellular permeability in response to bacterial signals, while occludin is essential for maintaining tight junction integrity, and is responsive to microbial population changes. *Salmonella* spp. upregulates the expression of the leaky protein claudin-2, adherent-invasive Escherichia coli (AIEC) disrupts tight junctions in Crohn’s disease patients, and probiotics like Bacillus subtilis and Bifidobacterium longum enhance tight junction integrity by upregulating claudin-1, occludin, and ZO-1, improving barrier function and reducing inflammation [43,44,45].

G-Protein Coupled Receptors (GPCRs): Receptors like GPR41 and GPR43 (FFAR2 and FFAR3), activated by SCFAs produced by bacterial fermentation of dietary fibers, and GPR119, which recognizes lipid-derived molecules from gut microbes, are crucial in mediating metabolic and immune responses [46,47,48]. GPR41 (FFAR3) is involved in energy regulation, influencing gut motility and energy extraction from the diet, and can modulate immune responses by affecting cytokine production, which is critical for managing inflammation. GPR43 (FFAR2) promotes anti-inflammatory responses; activation by SCFAs can suppress inflammatory cytokine production, help resolve inflammation, and maintain gut homeostasis by acting on various immune cells, including neutrophils and macrophages. Bacteroides thetaiotaomicron and *Faecalibacterium prausnitzii* activate GPR41 and GPR43 through short-chain fatty acids (SCFAs), such as propionate and butyrate, respectively, promoting anti-inflammatory effects and maintaining gut homeostasis [46,47,48].

Immune signaling molecules: NF-κB is a critical transcription factor mediating inflammatory responses to bacterial detection, the NF-κB pathway is activated by microbial infections, inflammatory cytokines, and cellular stress signals. Overall, the activation of NF-κB is associated with inflammation and is detrimental to the development of colorectal cancer (CRC). It is known that Helicobacter pylori activates the NF-κB pathway through IKK-mediated phosphorylation of IκB, Fusobacterium nucleatum activates NF-κB via the TLR4/MYD88 signaling pathway, and Lactobacillus casei inhibits NF-κB activation by decreasing the expression of NF-κB p65 and IκB, collectively affecting inflammation and cancer progression [49]. Interleukins such as TNF-α, IL-6, IL-10, and IL-22 play roles in regulating immune tolerance and epithelial barrier function in response to commensal bacteria. *Lactobacillus rhamnosus* and *Bifidobacterium breve* have been found to affect interleukin secretion by promoting anti-inflammatory cytokines such as IL-10 and reducing pro-inflammatory cytokines like IL-6 and TNF-α [50]. Chemokines like CCL20 and its receptor CCR6 are involved in recruiting immune cells to the gut mucosa in response to microbial signals. *Bacteroides fragilis* induces chemokine secretion through the activation of the NF-κB pathway, increasing the expression of chemokines like CXCL1, CXCL2, and CXCL5, while *Faecalibacterium prausnitzii* promotes the release of anti-inflammatory cytokine IL-10 and inhibits T cell proliferation in a CD39-dependent pathway [51].

Adhesion molecules and secretory molecules: E-cadherin maintains epithelial integrity and interacts with bacterial components. *Bacteroides fragilis* cleaves E-cadherin via its toxin, *Candida albicans* disrupts E-cadherin integrity, *Escherichia coli* displaces E-cadherin from adherens junctions, while *Saccharomyces boulardii* strengthens E-cadherin junctions [52]. Mucins (e.g., MUC2) are major components of the mucus layer, serving as a barrier and habitat for gut microbes, modulating their interaction with epithelial cells. Certain species, such as *Clostridium difficile* and *Helicobacter pylori*, affect mucin production by altering mucin composition and glycosylation patterns, while *Clostridium difficile* decreases MUC2 expression, leading to defective mucosal barrier function, and *Helicobacter pylori* impairs fucosylation of gastric MUC5AC, reducing mucus thickness and increasing permeability [53].

Other receptors and signaling molecules: Regenerating islet-derived protein 3 gamma (RegIIIγ) is an antimicrobial peptide upregulated in response to *Clostridia* spp. propionate signaling through IL-22 signaling [54]. C-type lectin receptors (CLRs) on host cells recognize bacterial components, influencing immune responses. The species of microorganisms that affect C-type lectin receptors include *Lactobacillus reuteri*, *Lactobacillus casei*, and *Lactobacillus acidophilus*, and these interactions involve the recognition of bacterial capsular polysaccharides and surface layer proteins which modulate immune responses and regulatory T-cell induction [55,56]. Retinoic acid-inducible gene I (RIG-1) and absent in melanoma 2 (AIM2) recognize microbial RNA and DNA, respectively, contributing to antiviral and antibacterial immune responses [57,58].

From the perspective of gut commensal bacteria, gene expression also plays a pivotal role in regulating interactions with the host immune system. The genes listed below can be classified based on their functions.

Adhesion and colonization factors: Genes such as fimH and spaP encode proteins that facilitate bacterial adherence to epithelial cells, promoting stable colonization of the gut and initiating immune system communication. Species of microorganisms such as *Escherichia coli*, particularly the adherent-invasive *E. coli* (AIEC) strain LF82, affect adhesion gene expression by expressing FimH, which binds to mannose on epithelial cells, triggering an inflammatory response in the gut [59,60].

Immune modulation and evasion: Genes involved in capsular polysaccharide synthesis, such as psaA, produce molecules that directly interact with the host immune system to promote tolerance and prevent inflammatory diseases. For example, polysaccharide A from *Bacteroides fragilis* drives the development of regulatory T cells [22].

Metabolic interactions: Genes like bile salt hydrolase (bsh) and the pdu operon are involved in metabolizing host-derived substances such as bile acids and contribute to vitamin synthesis and energy harvest. These metabolic products affect the host’s immune system by altering gut pH, influencing bile acid signaling, and providing metabolic signals that modulate immune responses. Species of microorganisms such as Lactobacillus, Bifidobacterium, Clostridium, and Enterococcus express bile salt hydrolases that deconjugate bile acids. Salmonella enterica uses the pdu operon by utilizing it for propanediol utilization, which is crucial for its survival and pathogenicity in the gut [61].

Toxin production and detoxification: Genes such as tdcG and hly are involved in toxin production, which can play roles in niche competition and immune stimulation, as seen in some strains of *Escherichia coli*. Conversely, detoxification genes help bacteria manage harmful compounds, potentially reducing gut inflammation [62,63,64].

Signal molecule production: Genes such as luxS and spxB are involved in the synthesis of signaling molecules like autoinducers and hydrogen peroxide. These molecules influence bacterial community behavior and host immune responses, facilitating communication within the microbiota and with host cells. Lactobacillus paraplantarum uses the luxS gene to produce autoinducer-2 (AI-2), promoting stress resistance, biofilm formation, and altering gene expression related to transporters and membrane proteins. *Streptococcus sanguinis* and *Streptococcus gordonii* use the spxB gene to produce hydrogen peroxide, which enhances their competitive fitness in the biofilm environment. Additionally, the *p40* gene product activates the epidermal growth factor receptor (EGFR) pathway in intestinal epithelial cells. *Lacticaseibacillus rhamnosus* uses the p40 gene product to activate the EGFR pathway by stimulating the release of ligands like heparin-binding EGF (HB-EGF) through the activation of ADAM17, which subsequently protects intestinal epithelial cells by preserving tight junctions, promoting mucin production, and preventing apoptosis [65,66,67,68].

As we observe, numerous studies highlight the exceptional importance of the production of short-chain fatty acids (SCFAs) by the gut microbiota for evaluating the effectiveness of checkpoint immunotherapy [69]. Furthermore, the SCFAs produced by the gut microbiome in the blood may serve as biomarkers for colorectal cancer [70]. The role of microbiota in the response to immunotherapy will be discussed in more detail in Chapter 6.

In developing potential tests to assess the effects of immunotherapy, one approach could be to determine the SCFA potential of the microbiome through quantitative and qualitative analysis of genes encoding the production of these SCFAs, mediated by specific bacterial genes within the gut microbiome. For instance, butyrate production relies on the activity of enzymes such as butyryl-CoA CoA transferase and butyrate kinase, predominantly found in bacterial genera like Faecalibacterium, Eubacterium, and Roseburia [71]. Acetate, another crucial SCFA, is produced through enzymes like acetate kinase and phosphate acetyltransferase, with Akkermansia and certain Bifidobacteria playing a central role [72]. Propionate production involves pathways such as the propanediol and succinate pathways, facilitated by enzymes like lactoyl-CoA dehydratase and methylmalonyl-CoA decarboxylase, found in Propionibacterium and Veillonella [69,73].

Advancements in whole-genome metagenomic sequencing have enabled researchers to delve deeply into the genetic material of the gut microbiota, identifying genes linked to SCFA production. This comprehensive data allows for the quantification of SCFA-related genes and their relative abundance. Bioinformatics tools such as MetaPhlAn, HUMAnN2, and the KEGG databases are instrumental in mapping these sequencing data to specific metabolic pathways, thus linking microbial community composition to functional capabilities like SCFA production [74,75].

Beyond genetic potential, metabolomics provides a direct measure of metabolic output, such as SCFA concentrations in fecal samples. Techniques like gas chromatography–mass spectrometry (GC-MS) are employed to quantify these metabolites, offering a functional validation of the microbiome’s activity as inferred from genomic data [76]. A holistic approach that integrates metagenomic and metabolomic data with clinical metadata, dietary information, and other relevant factors is crucial for a comprehensive assessment of the microbiome’s role in health and disease. This integration helps elucidate the complex interactions between diet, microbiome-derived SCFAs, and their impact on the host, including responses to treatments like immunotherapy.

In conclusion, the molecular mechanisms encoded by both host and microbial genes play pivotal roles in shaping the interactions between the gut microbiome and the immune system. These mechanisms regulate immune responses, maintain epithelial barrier integrity, and facilitate communication, underscoring the complexity and significance of microbiome–immune system interplay.

The main mechanisms of interaction between the microbiome and innate immunity described in the chapter are shown in Figure 1.

Bacterial flora can exhibit both pro-inflammatory and anti-inflammatory properties depending on the species. A typical representative of the former is *Fusobacterium nucleatum*, while the latter is exemplified by *Bifidobacterium* spp. An important aspect of the flora’s impact on immunity is the site of action. In the mucosal layer of the gut lumen, bacteria can form biofilms, where biofilm formation factors such as fimH and spaP enhance the potential of the dominant species.

The integrity of the epithelial barrier is a crucial component of antimicrobial defense. Pro-inflammatory bacteria, such as *Fusobacterium nucleatum* and *Escherichia coli*, disrupt tight junction molecules (claudins and occludins) and E-cadherin, increasing barrier permeability through toxin production and activation of inflammatory pathways (NF-kB, IL-1β, IL-6, and TNF-α). In contrast, anti-inflammatory bacteria, such as *Faecalibacterium prausnitzii* and *Akkermansia muciniphila*, strengthen these molecules and maintain barrier integrity by producing butyrate and stimulating the production of anti-inflammatory cytokines (IL-10) and mucins.

In the subepithelial zone, the interaction of bacterial factors with immune cells (macrophages, dendritic cells, granulocytes, lymphocyte subpopulations) is most pronounced. Short-chain fatty acids (SCFAs), such as butyrate, produced by bacteria, possess anti-inflammatory properties through interaction with G-Protein Coupled Receptors (GPCRs), suppressing the production of inflammatory cytokines (IL-1β, IL-6, and TNF-α) and stimulating the secretion of anti-inflammatory cytokines (IL-10). Pro-inflammatory effects are mediated through the NF-κB signaling pathway. Toll-like receptors (TLR4 and TLR2) recognize microbial lipopolysaccharides and lipoproteins, activating signaling pathways that lead to the production of pro-inflammatory cytokines and chemokines. NOD receptors (NOD1 and NOD2) recognize peptidoglycan fragments of bacterial cell walls, activating signaling pathways and the production of pro-inflammatory cytokines. These mechanisms provide protection to the host but, in cases of dysbiosis, can contribute to carcinogenesis.

## 5. Microbiome and Colorectal Cancer Immunity

The role of the gut microbiota in shaping immunity is crucial from early childhood, marked by significant variability in the interactions between the microbiota and the immune system. The primary factors influencing these interactions have been elucidated. In CRC, the complexity of these dynamics increases due to the tumor’s impact on immune responses. Tumors express neoantigens and evolve mechanisms to evade immune detection, which, in turn, alters the gut microbiota composition.

The immune response in CRC is notably influenced by the tumor’s neoantigen burden. This is particularly characteristic of tumors with a deficiency in the mismatch repair system, manifesting as microsatellite instability [77]. Furthermore, a subset of CRC exhibits evasion of immune surveillance mediated by checkpoint regulators of immunity [78]. The immune response in CRC is also marked by the infiltration of the stroma by lymphocytes and tumor-associated macrophages [79]. Additionally, the immune response in cancer is accompanied by changes in the pattern of gene expression, which directly or indirectly regulate the immune response [80]. All these aspects may be subjects of tumor–microbiome interactions. The main factors involved in anti-tumor immunity are presented in Figure 2.

The main factors involved in anti-tumor immunity are presented in Figure 2.

(1) Antigen release and presentation: Colorectal cancer cells undergo necrosis or apoptosis, leading to the release of tumor-associated antigens (TAAs). Dendritic cells (DCs) capture these TAAs and process them into peptide fragments. The processed antigens are then presented on the surface of DCs via major histocompatibility complex (MHC) molecules (MHC class I and II). (2) Dendritic Cell Activation and Migration: Upon antigen uptake, DCs become activated by signals from the tumor microenvironment, such as damage-associated molecular patterns (DAMPs) and pathogen-associated molecular patterns (PAMPs). Activated DCs migrate to regional lymph nodes, where they present the processed antigens to naïve T cells. (3) T cell priming and activation: Naïve T cells recognize the antigens presented by DCs through their T cell receptors (TCRs) in the context of MHC molecules. Effective T cell activation requires additional co-stimulatory signals provided by DCs, such as CD80/CD86 binding to CD28 on T cells. Upon activation, naïve T cells differentiate into effector T cells, including cytotoxic T lymphocytes (CTLs) and helper T cells (Th1). (4) Clonal expansion and trafficking: Activated T cells undergo clonal expansion to generate a large pool of effector cells. Effector T cells exit the lymph nodes and travel through the bloodstream to the tumor site, guided by chemokine gradients. (5) Infiltration and tumor cell recognition: Effector T cells infiltrate the tumor microenvironment. CTLs recognize cancer cells presenting TAAs via MHC class I molecules, involving TCR binding to the antigen-MHC complex. (6) Effector function and tumor cell killing: CTLs exert their effector functions by releasing perforin and granzymes, which induce apoptosis in the targeted cancer cells. Another mechanism involves the Fas/FasL pathway, where the Fas ligand (FasL) on CTLs binds to the Fas receptor on tumor cells, triggering apoptosis. (7) Sustaining the immune response: Some activated T cells differentiate into memory T cells, providing long-lasting immunity and the ability to respond more rapidly upon re-exposure to the same antigens. The immune response is tightly regulated by immune checkpoints (e.g., PD-1/PD-L1 and CTLA-4) to prevent excessive tissue damage and maintain homeostasis. (8) Modulation by the tumor microenvironment: A successful anti-tumor response must overcome various immunosuppressive factors in the tumor microenvironment, such as regulatory T cells (Tregs), myeloid-derived suppressor cells (MDSCs), and immunosuppressive cytokines (e.g., TGF-β, IL-10). (9) Modulation by microbes: Microbial metabolites, such as short-chain fatty acids (SCFAs), influence the activation of CD8+ T cells. For instance, butyrate enhances CD8+ T cell responses through the ID2-dependent IL-12 signaling pathway, which is critical for anti-tumor immune response. Studies have shown that microbial metabolites, such as butyrate and pentanoate, enhance T cell function by inhibiting the activity of class I histone deacetylases (HDAC), increasing mTOR activity, and boosting the production of effector molecules like CD25, IFN-γ, and TNF-α in CTLs and CAR-T cells. The gut microbiota, through its metabolites and interactions with immune cells, can modulate the functions of effector CD8+ T cells. SCFAs can modulate cytokine expression and metabolic pathways, affecting CTL functions. Microbial metabolites, such as butyrate, enhance the metabolism of CD8+ T cells, which is important for their differentiation into memory cells. Butyrate stimulates the metabolic reprogramming of CD8+ T cells, including the uncoupling of the Krebs cycle from glycolytic input. This promotes the utilization of glutamine and fatty acid oxidation for oxidative phosphorylation, which is crucial for the transition of CD8+ T cells into memory cells.

Microsatellite instability (MSI) in CRC is characterized by high levels of MSI (MSI-H), which are associated with numerous mutations that generate neoantigens, enhancing immunogenicity. Studies show that MSI-H CRCs typically exhibit a dense infiltration of cytotoxic T cells and are linked with improved responses to immunotherapy [81]. It is hypothesized based on existing studies that the activation of the immune system by tumors with MSI may shift the balance of the microbiota.

The dependency of the microbiome on microsatellite instability and mutational burden is a significant area of investigation. Intratumoral microbes are reported to have multifunctional roles in carcinogenesis, with MSI being associated with enhanced tumor immunity and an increased mutational burden. Research utilizing whole transcriptome and whole genome sequencing to analyze microbial abundance data has demonstrated associations of intratumoral microbes with MSI, survival rates, and MSI-relevant tumor molecular characteristics across multiple cancer types, including CRC, stomach adenocarcinoma, and endometrial carcinoma. Notably, strong associations were observed between multiple CRC-associated genera, such as Dialister and Casatella, and MSI status. The abundance of Dialister and Casatella was correlated with improved overall survival (hazard ratio for mortality = 0.56, 95% confidence interval = 0.34 to 0.92; and hazard ratio for mortality = 0.44, 95% confidence interval = 0.27 to 0.72, respectively), particularly when comparing higher to lower quantiles. Additionally, multiple intratumor microbes were associated with immune genes and tumor mutational burden. The diversity of oral cavity-originating microbes was also linked to MSI among patients with CRC and stomach adenocarcinoma [81].

In terms of bacteria associated with deficient mismatch repair (dMMR), *Fusobacterium nucleatum* and *Fusobacterium periodonticum* have been significantly enriched in dMMR CRC. These bacteria are implicated in colorectal carcinogenesis and might influence the tumor microenvironment through mechanisms that could interfere with DNA repair systems [82] Similarly, *Bacteroides fragilis*, also enriched in dMMR CRC, has been implicated in inflammation-driven CRC development and progression, potentially affecting the immune response and further destabilizing the genomic integrity of tumor cells [83].

Immune Checkpoints: Proteins such as PD-1, PD-L1, and CTLA-4 are frequently upregulated in CRC. The expression of PD-L1 on tumor cells and tumor-associated immune cells can inhibit anti-tumor immunity by engaging with PD-1 on T cells. Therapies targeting these checkpoints have demonstrated promise in enhancing the immune response against tumors.

Recent studies have defined the response to PD-1/PD-L1 checkpoint immunotherapy in gastrointestinal cancer patients, including CRC, based on clinical outcomes observed post-treatment. Researchers utilized 16S and shotgun sequencing to explore the gut methagenome. A good response is associated with an increased Prevotella/Bacteroides ratio—patients with a higher ratio of Prevotella to Bacteroides in their gut microbiome tended to exhibit better responses to PD-1/PD-L1 therapy. Specific microbial taxa associated with a positive response include Prevotella and Ruminococcaceae, which were found in higher abundance in responders, suggesting a potentially beneficial role in modulating the immune response favorable to PD-1/PD-L1 blockade. Lachnospiraceae, another microbial group, was also more abundant in responders, indicating its potential involvement in enhancing immunotherapy efficacy [69].

The study also highlights that certain metabolic pathways influenced by the gut microbiome, such as those related to short-chain fatty acid (SCFA) production, might impact the response to immunotherapy. SCFA-producing bacteria like Eubacterium, Lactobacillus, and Streptococcus were positively associated with responses to PD-1/PD-L1 inhibitors [69,84,85].

Conversely, microbial taxa associated with a poor response identified Bacteroides as more abundant in non-responders, suggesting a possible negative impact on the effectiveness of PD-1/PD-L1 therapies [69].

Tumor-infiltrating lymphocytes (TILs): The presence and composition of TILs in CRC tumors serve as significant prognostic markers. High densities of CD8+ cytotoxic T lymphocytes, for instance, are associated with better survival outcomes. In contrast, regulatory T cells (Tregs), which are marked by FOXP3 expression, can suppress anti-tumor immune responses and correlate with poorer outcomes. Research has shown that specific gut bacteria such as *Fusobacterium nucleatum* engage immune signaling pathways, including NF-κB, via TLR/MyD88 signaling in tumor-infiltrating immune cells. This engagement not only promotes the expression of pro-inflammatory cytokines and vascular endothelial growth factor (VEGF), which in turn facilitates angiogenesis within the tumor microenvironment (TME), but *Fusobacterium nucleatum* is also generally associated with worse clinical outcomes. Interestingly, it can evoke the expression of chemokines that recruit T cells to the tumor site, adding complexity to its role in CRC progression [86]. Similarly, *Bacteroides fragilis* and *Escherichia coli* have been found to stimulate chemokine production that favors the recruitment of beneficial T cells into tumor tissues. This response is likely mediated through the bacteria’s interaction with pattern recognition receptors on tumor or immune cells, leading to an enhanced immune response [87,88]. These mechanisms contribute significantly to the development and progression of CRC, underscoring the complex interplay between microbiota and tumor immune dynamics.

Cytokine profiles: The balance of pro-inflammatory and anti-inflammatory cytokines within the tumor microenvironment significantly influences CRC progression. High levels of pro-inflammatory cytokines such as IL-6 and TNF-α can promote tumor growth and metastasis. Conversely, cytokines like IFN-γ are generally known to enhance anti-tumor immunity [89]. It has been discovered that *Ruminococcus gnavus*, Proteobacteria, *Escherichia coli,* and *Lachnospiraceae bacteria* are associated with diets rich in animal proteins and fats and are implicated in promoting pro-inflammatory states, linked with increased gut permeability and inflammation [90]. On the other side, *Faecalibacterium prausnitzii* and Roseburia produce short-chain fatty acids (SCFAs) like butyrate, which are known for their anti-inflammatory effects on the gut lining. Bifidobacterium, associated with diets high in fiber (plant-based foods), plays a beneficial role in strengthening the gut barrier and reducing inflammation. Lactic acid bacteria, found in higher abundance in fermented dairy products like yogurt, are linked to anti-inflammatory properties in the gut [90].

Immune-related gene signatures: Advances in genomic profiling have enabled the development of gene signatures that reflect the immune contexture of tumors. These signatures potentially predict responses to immunotherapy and overall survival [91]. Associations between specific microbial taxa and host gene expression changes in CRC patients highlight the implications of these changes:

*Fusobacterium nucleatum*: Increased abundance of *Fusobacterium nucleatum* is associated with the downregulation of genes such as Peptidase Inhibitor 16 (PI16), Fc Receptor-Like A (FCRLA), and Lymphocyte Specific Protein 1 (LSP1), including the downregulation of TNFSF9 gene expression. Generally, the presence of *Fusobacterium nucleatum* negatively affects CRC by promoting inflammation and cancer progression. The downregulation of these genes may contribute to immune evasion by the tumor, potentially worsening patient outcomes. Specifically, the downregulation of TNFSF9 by *Fusobacterium nucleatum* could modify the tumor immune environment, facilitating tumor progression by impairing the immune response [92,93].

*Prevotella spp*.: A decreased abundance of Prevotella 2 correlates with the downregulated expression of Metallothionein 1M (MT1M), a gene involved in metal ion metabolism and protection against oxidative stress. Its downregulation may disrupt cellular defense mechanisms against cancer progression, suggesting a potentially negative effect on CRC [92].

Halomonadaceae: An increase in Halomonadaceae is associated with decreased expression of Reelin (RELN), a gene involved in cell adhesion, migration, and neuronal transmission. Its downregulation may contribute to enhanced tumor invasiveness and poor prognosis in CRC, indicating a likely negative impact [92].

*Paeniclostridium spp*.: A decreased abundance correlates with upregulated expression of *Phospholipase C Beta 1* (*PLCB1*), a gene involved in various cellular processes including proliferation and apoptosis. The specific impact of increased *PLCB1* expression in CRC remains unclear and may depend on other cellular signaling pathways, making its effect ambiguous [92].

*Enterococcus spp*.: A decrease in Enterococcus levels is linked to the downregulation of *Immunoglobulin Superfamily Member 9* (IGSF9), a gene involved in immune cell signaling. Its downregulation might impair immune surveillance, potentially facilitating tumor progression, and suggesting a potentially negative effect on CRC [92].

*Bacteroides fragilis*: This bacterium is associated with changes in metabolic pathways related to the one-carbon pool by folate, affecting nucleotide synthesis, amino acid metabolism, and methylation processes. While *Bacteroides fragilis* is known for its pro-inflammatory effects, which can be detrimental by promoting CRC progression, it also plays roles in normal gut physiology and metabolism that could be potentially harnessed for therapeutic benefits, rendering its effect ambiguous [93].

Tumor-associated macrophages: Tumor-associated macrophages (TAMs) significantly influence CRC progression, with their role shifting between pro-tumoral (M2) and anti-tumoral (M1) activities. The presence of M2 macrophages is often associated with tumor progression and poor prognosis, highlighting the critical nature of macrophage polarization in cancer outcomes [94]. An imbalance in the gut microbiota, such as an overabundance of *Escherichia coli*, can drive TAMs towards an M2 polarization via TLR4-dependent mechanisms. This polarization stimulates tumor cells to produce enzymes like cathepsin K, promoting tumor growth [82].

On the other hand, Bifidobacterium adolescentis has been shown to have a positive impact on macrophage behavior, utilizing TLR2 signaling to encourage the development of macrophages that express Decorin. This protein helps suppress colorectal carcinogenesis by inhibiting tumor growth [95], demonstrating the potential benefits of manipulating gut bacteria to favor anti-tumoral immune responses.

*Fusobacterium nucleatum* plays a complex role in the tumor microenvironment. Although generally associated with negative CRC outcomes, it can induce M1 polarization through the microbial metabolite trimethylamine N-oxide, highlighting the dual influence of these bacteria [96].

Additionally, targeting the complement component C5aR1 has emerged as a viable strategy to promote M1 polarization of TAMs. This process is mediated by AKT2-driven stabilization of the 6-phosphofructokinase muscle type (PFKM), which is influenced by the gut microbiota through IL-22-regulated antimicrobial peptides. These peptides modulate the microbiota and trigger the release of flagellin, which supports the M1 polarization enhancing the immune system’s capability to combat tumor cells [96].

In conclusion, the interplay between the gut microbiota and immune mechanisms within the CRC environment underscores the profound impact of microbial dynamics on tumor progression and response to therapy. Research revealing how microbes such as *Fusobacterium nucleatum* and *Bacteroides fragilis* influence gene expression and immune cell behavior highlights the potential of targeting these interactions to enhance immunotherapy efficacy. This growing understanding promises new therapeutic strategies aimed at modulating the microbiome to improve CRC treatment outcomes. *Prevotella spp*.: A decreased abundance of Prevotella 2 correlates with the downregulated expression of Metallothionein 1M (MT1M), a gene involved in metal ion metabolism and protection against oxidative stress. Its downregulation may disrupt cellular defense mechanisms against cancer progression, suggesting a potentially negative effect on CRC [97].

Halomonadaceae: An increase in Halomonadaceae is associated with decreased expression of *Reelin* (*RELN*), a gene involved in cell adhesion, migration, and neuronal transmission. Its downregulation may contribute to enhanced tumor invasiveness and poor prognosis in CRC, indicating a likely negative impact [97].

*Paeniclostridium spp*.: A decreased abundance correlates with upregulated expression of *Phospholipase C Beta 1* (*PLCB1*), a gene involved in various cellular processes including proliferation and apoptosis. The specific impact of increased *PLCB1* expression in CRC remains unclear and may depend on other cellular signaling pathways, making its effect ambiguous [97].

*Enterococcus* spp.: A decrease in Enterococcus levels is linked to the downregulation of *Immunoglobulin Superfamily Member 9* (*IGSF9*), a gene involved in immune cell signaling. Its downregulation might impair immune surveillance, potentially facilitating tumor progression and suggesting a potentially negative effect on CRC [97].

*Bacteroides fragilis*: This bacterium is associated with changes in metabolic pathways related to the one-carbon pool by folate, affecting nucleotide synthesis, amino acid metabolism, and methylation processes. While Bacteroides fragilis is known for its pro-inflammatory effects, which can be detrimental by promoting CRC progression, it also plays roles in normal gut physiology and metabolism that could be potentially harnessed for therapeutic benefits, rendering its effect ambiguous [98].

Tumor-associated macrophages: Tumor-associated macrophages (TAMs) significantly influence colorectal cancer (CRC) progression, with their role shifting between pro-tumoral (M2) and anti-tumoral (M1) activities. The presence of M2 macrophages is often associated with tumor progression and poor prognosis, highlighting the critical nature of macrophage polarization in cancer outcomes [99]. An imbalance in the gut microbiota, such as an overabundance of *Escherichia coli*, can drive TAMs towards an M2 polarization via TLR4-dependent mechanisms. This polarization stimulates tumor cells to produce enzymes like cathepsin K, promoting tumor growth [100].

On the other hand, Bifidobacterium adolescentis has been shown to have a positive impact on macrophage behavior, utilizing TLR2 signaling to encourage the development of macrophages that express Decorin. This protein helps suppress colorectal carcinogenesis by inhibiting tumor growth [101], demonstrating the potential benefits of manipulating gut bacteria to favor anti-tumoral immune responses.

*Fusobacterium nucleatum* plays a complex role in the tumor microenvironment. Although generally associated with negative CRC outcomes, it can induce M1 polarization through the microbial metabolite trimethylamine N-oxide, highlighting the dual influence of these bacteria [102].

Additionally, targeting the complement component C5aR1 has emerged as a viable strategy to promote M1 polarization of TAMs. This process is mediated by the AKT2-driven stabilization of the 6-phosphofructokinase muscle type (PFKM), which is influenced by the gut microbiota through IL-22-regulated antimicrobial peptides. These peptides modulate the microbiota and trigger the release of flagellin, which supports the M1 polarization enhancing the immune system’s capability to combat tumor cells [102].

In conclusion, the interplay between the gut microbiota and immune mechanisms within the colorectal cancer (CRC) environment underscores the profound impact of microbial dynamics on tumor progression and response to therapy. Research revealing how microbes such as *Fusobacterium nucleatum* and *Bacteroides fragilis* influence gene expression and immune cell behavior highlights the potential of targeting these interactions to enhance immunotherapy efficacy. This growing understanding promises new therapeutic strategies aimed at modulating the microbiome to improve CRC treatment outcomes.

## 6. The Influence of Gut Microbiota on Immunotherapy Efficacy in Colorectal Cancer

The gut microbiota profoundly influences the efficacy of immunotherapy in CRC by modulating the host immune response through complex microbial-host interactions. Emerging evidence suggests that specific commensal bacteria can enhance anti-tumor immunity by promoting the maturation and activation of dendritic cells, augmenting the infiltration of cytotoxic T lymphocytes into the tumor microenvironment, and producing metabolites that directly influence immune cell function.

Currently, the most relevant therapy for CRC is immune checkpoint inhibitor therapy, while other types of immunotherapies—such as adoptive T-cell therapy and CAR-T cell therapy—are still in the research stage. Immune checkpoint inhibitor therapy is a form of cancer treatment that blocks receptors and their ligands, such as PD-1, PD-L1, and CTLA-4, which regulate immune responses, thereby enhancing the ability of T cells to recognize and attack tumor cells.

Researchers have noted several common mechanisms associated with the microbiota that potentially influence the efficacy of attempts to impact anti-tumor immunity. For example, one study explored how alterations in the gut microbiome affect T cell receptor (TCR) repertoires in CRC patients, which is pertinent to immunotherapy effectiveness. The TCR repertoires in CRC patients exhibit high clonality and low diversity, indicating an immune response focused on a limited set of antigens. This effect correlated with increased levels of *Fusobacterium nucleatum*, *Escherichia coli*, and Prevotella, and decreased levels of *Faecalibacterium prausnitzii* and Roseburia. The carcinogenic action of bacteria is linked to activating the nuclear factor-κB (NF-κB) pathway and promoting myeloid cell infiltration, leading to a pro-inflammatory and immunosuppressive tumor microenvironment. It upregulates cytokines like IL-6 and TNF-α and downregulates miRNA18a via the TLR-4/MYD88 pathway, further supporting tumor growth and immune evasion. The study highlighted that gut microbiota influences T cell populations, particularly regulatory T cells (Tregs) [103].

As an example of how gut flora can exert systemic effects through microbial metabolites, a study involved 50 patients with metastatic melanoma. Additionally, mice models were used to investigate the effects of SCFAs on CTLA-4 blockade. The study discovered that a higher abundance of *Faecalibacterium prausnitzii* and other Firmicutes was associated with a better clinical outcome, while higher levels of Bacteroides were linked to poor clinical outcomes. High blood levels of the SCFAs butyrate and propionate were associated with resistance to CTLA-4 blockade, reducing the efficacy of immunotherapy by upregulating CD80/CD86 on dendritic cells, ICOS (Inducible T-cell COStimulator) gene expression in T cells, and the accumulation of tumor-specific T cells and memory T cells. This suggests that butyrate limits the immune response induced by CTLA-4 blockade [104].

The systemic impact of gut flora on immune regulation and response to immunotherapy can be seen in studies of tumors outside the gut. For example, research investigated the role of gut microbiome diversity and specific composition in the efficiency of immune checkpoint inhibitor (ICI) therapy in 28 non-small cell lung cancer (NSCLC) patients. Evaluations included progression-free survival (PFS), Immune-related adverse events (irAE), and microbiota diversity and composition (alpha and beta diversity indices). The study discovered that the *Blautia* genus was enriched in responders and associated with longer PFS. The presence of specific microbial taxa was linked to the enrichment of immune markers like PD-L1 expression and T-cell activation. The study concludes that gut microbiota diversity is strongly associated with the response to ICI therapy in cancer patients [105].

Another example is the influence of the gut bacterium *Parabacteroides distasonis* on the efficacy of anti-PD-1 immunotherapy in bladder cancer. It was found that the levels of *Parabacteroides distasonis* in the feces of healthy individuals were significantly higher than those in bladder cancer patients. Gene Set Enrichment Analysis (GSEA) showed significant upregulation in pathways related to the anti-tumor immune response in the *P. distasonis* and anti-PD-1 mAb treatment group, including Natural Killer (NK) Cell-Mediated Cytotoxicity, the T Cell Receptor Signaling Pathway, Cell Receptor Signaling Pathway, Cytokine-Cytokine Receptor Interaction, and the Chemokine Signaling Pathway [106].

One unique feature of the gut flora is its interaction with bile components. A study examined 88 patients with extrahepatic cholangiocarcinoma, intrahepatic cholangiocarcinoma, and gallbladder cancer who received PD-1/PD-L1 inhibitors. Patients were classified into durable clinical benefit and non-durable clinical benefit groups based on their responses and the duration of stable disease. The study found that the *Alistipes* genus was positively correlated with survival and enriched in the durable clinical benefit group, whereas the Bacillus class and particularly the Lactobacillales order were negatively correlated with survival and enriched in the non-durable clinical benefit group. Metabolites impacted the efficacy of immunotherapy in patients. Pyrrolidine was negatively correlated with survival, while 4-[(hydroxymethyl)nitrosoamino]-1-(3-pyridinyl)-1-butanone was positively correlated with survival. The study found that the gut microbiome and its metabolites were involved in several KEGG pathways, including glycine, serine, and threonine metabolism [107].

Numerous recent studies highlight the relevance and importance of the species composition, diversity, and properties of the gut microbiome concerning the efficacy of immune checkpoint inhibitor therapy CRC. Several molecular and cellular mechanisms are involved in regulating the response to immunotherapy. A study focused on the general concept of immunotherapy efficacy as measured by immunoscores (IS) based on the density of CD3+ and CD8+ tumor-infiltrating lymphocytes (TILs) in the center of the tumor and the invasive margin. The study examined 94 patients with CRC and discovered that a high abundance of the Lachnospiraceae family in the gut microbiome was associated with high immunoscores, suggesting a positive effect on the efficacy of immunotherapy. This effect is linked to the Lachnospiraceae family producing butyric acid [108].

Another study explored the role of the gut microbiome, specifically *Lactobacillus gallinarum*, in enhancing the efficacy of anti-PD1 immunotherapy CRC. The study used mouse models, including syngeneic mouse models and an azoxymethane/dextran sulfate sodium (AOM/DSS)-induced CRC, to evaluate tumor size, tumor weight, and tumor volume. Additionally, immune cell infiltration and effector function, particularly CD8+ T cells and regulatory T cells (Tregs), were evaluated. The study discovered that *Lactobacillus gallinarum* improved the efficacy of anti-PD1 therapy by reducing intratumoral infiltration of Tregs and enhancing the effector function of CD8+ T cells. *Lactobacillus gallinarum*-derived indole-3-carboxylic acid (ICA) was identified as a functional metabolite that improved anti-PD1 efficacy by suppressing IDO1 expression, reducing kynurenine production, thereby inhibiting Treg differentiation and enhancing CD8+ T cell function [84].

Another study on mouse models, including the MC38 colon carcinoma model, investigated the mechanisms by which gut microbiota influences the efficacy of anti-PD-1 and anti-PD-L1 therapies. Stool samples from melanoma patients (one responder and two non-responders to anti-PD-1 therapy) were used for fecal microbiota transplantation experiments in mice. The study identified that *Coprobacillus cateniformis* and *Erysipelatoclostridium ramosum* enhanced the efficacy of anti-PD-L1 immunotherapy by downregulating PD-L2 expression on dendritic cells in lymph nodes, a crucial role in promoting anti-tumor immunity. PD-L2 suppression was mediated through interactions with Repulsive Guidance Molecule b (RGMb), which, when blocked, enhanced the efficacy of PD-1/PD-L1 inhibitors [109].

Another study on animal models evaluated the effect of the microbiota on another immune checkpoint molecule. The study investigated how lysates of Lactobacillus acidophilus combined with CTLA-4-blocking antibodies (CTLA-4 IgG (clone# 9H10)) enhanced anti-tumor immunity in a BALB/c mouse model of colon cancer. The enhanced efficacy of the combined treatment was associated with reduced tumor formation, increased CD8+ T cells, increased effector memory T cells, decreased Treg cells, and decreased M2 macrophages in the tumor microenvironment. Additionally, the lysates inhibited M2 polarization and reduced IL-10 levels in macrophages, suppressing the growth of Proteobacteria [110].

Research on the effects of changes in the gut microbiota and their impact on the efficacy of anti-PD-1 therapy is actively conducted on mouse models. One such study investigated how fecal microbiota transplantation (FMT) can enhance the efficacy of anti-PD-1 therapy (PD-1 antibody RMP1-14) in treating CRC. It was an animal study using mouse models to investigate the effects of FMT from healthy mice on anti-PD-1 therapy. The study discovered that beneficial species such as *Bacteroides thetaiotaomicron*, *B. fragilis*, and *B. cellulosilyticus* were increased with FMT, contributing to improved anti-PD-1 therapy efficacy. Detrimental species like *Bacteroides ovatus* and *Lactobacillus murinus* were decreased with FMT, contributing to the enhanced therapeutic effect. The role of the metabolite punicic acid, which was upregulated after FMT and promoted the response to anti-PD-1 therapy due to its immunomodulatory functions, was also identified [111].

Other researchers investigated the role of the gut microbiome, particularly focusing on specific bacterial species and their metabolites, in enhancing the efficacy of anti-PD-L1 antibodies immunotherapy on mouse models of CRC. The study discovered that *Bifidobacterium pseudolongum*, *Lactobacillus johnsonii*, and *Olsenella* species significantly enhanced the efficacy of anti-PD-L1 therapy, promoting anti-tumor immunity. The study identified the metabolite inosine, produced by *Bifidobacterium pseudolongum*, as a key factor that enhances the efficacy of the therapy. Inosine enhances Th1 differentiation and anti-tumor immunity via the A2A receptor signaling pathway on T cells, requiring costimulation and IL-12 receptor engagement [112].

In human CRC, research involving patients is of great interest. One such study examined how the gut microbiome affects the efficacy of the combination therapy of regorafenib (a multikinase inhibitor) and toripalimab (a recombinant, humanized immunoglobulin G4 (IgG4) monoclonal antibody against PD-1) in metastatic CRC patients. Patients were divided into responders and non-responders based on their clinical response to immunotherapy. The study examined 42 patients and found that increased levels of the genus Fusobacterium were found in non-responders compared to responders, correlating with shorter progression-free survival and indicating a negative correlation with the efficacy of the combination therapy [113].

Another large study, including 800 patients with tumors of different locations, evaluated both bacterial and fungal components in the efficiency of immune checkpoint inhibitor (ICI) therapy in humans. The immunotherapies investigated included anti-PD-1 (pembrolizumab and nivolumab), anti-PD-L1 (atezolizumab and durvalumab), and anti-CTLA-4 (ipilimumab) therapies. The study evaluated Objective response rate (ORR), progression-free survival (PFS), overall survival (OS), and immune cell infiltration, specifically the enrichment of exhausted T cells and the expression of inhibitory receptors. The study discovered that the fungal species *Schizosaccharomyces octosporus* was enriched in responders and contributed positively to the efficacy of ICI therapy. In contrast, the fungal species *Pseudocercospora musae* were enriched in non-responders, indicating a negative impact on therapy response. *Schizosaccharomyces octosporus* may ferment starch into short-chain fatty acids (SCFAs) in responders. The study also revealed that the amount of these fungal species in the feces correlated with the exhausted T cell signature, including genes typically upregulated in exhausted T cells, such as those encoding inhibitory receptors PD-1 and CTLA-4. The exhausted T cell signature is associated with the response to ICI therapy [114].

Another research investigated how the gut microbiome affects clinical responses to anti-PD-1/PD-L1 immunotherapy (nivolumab, pembrolizumab, atezolizumab, and durvalumab) in gastrointestinal cancer patients, including CRC, esophageal cancer, and gastric cancer. The study examined 74 patients with advanced-stage cancer. The main criteria for evaluating the efficiency of immunotherapy included progression-free survival (PFS), clinical response (partial response, stable disease, and progressive disease) according to the RECIST v1.1 standard, and microbiota composition and diversity (Prevotella/Bacteroides ratio). The study discovered that a higher abundance of Prevotella, Ruminococcaceae, and Lachnospiraceae was associated with better clinical responses. Conversely, a higher abundance of Bacteroides was negatively associated with PFS. The study found that gut bacteria capable of producing short-chain fatty acids (SCFAs) like Eubacterium, Lactobacillus, and Streptococcus were positively associated with anti-PD-1/PD-L1 response [69].

Another small clinical study investigated how gut microbiota affects the anti-tumor activity of the combination therapy with cetuximab (an anti-EGFR monoclonal antibody) and avelumab (an anti-PD-L1 monoclonal antibody) in patients with metastatic CRC. The study examined a total of 24 patients, specifically 14 patients with metastatic CRC. The study discovered that Agathobacter and Blautia were found to be enriched in long-term responders with better PFS in mCRC patients. The primary factor influencing immunotherapy was suggested to be the production of the metabolite butyrate [115].

Another small study involving 72 patients with advanced cancers of various locations, including CRC, investigated how the composition of the gut microbiota is associated with the clinical response to anti-PD-1 immunotherapy. The responders before immunotherapy had significantly enriched Archaea, Lentisphaerae, Victivallaceae, Methanobacteriaceae, Methanobacteria, Euryarchaeota, Methanobrevibacter, and Methanobacteriales. Non-responders had significantly enriched Clostridiaceae. The study suggested that specific bacterial families and their functional pathways might support a favorable immune response but did not detail specific molecular or cellular mechanisms of interaction between microbiome species and immunotherapy [116].

Another study investigates how the dysregulation of innate lymphoid cells (ILC3s) and their interaction with gut microbiota affect anti-PD-1 immunotherapy resistance in CRC. The study involved 72 human subjects with CRC and used mouse models to investigate the mechanisms and effects observed in human samples. The study discovered that an increased abundance of the genus Bacteroides was associated with resistance to anti-PD-1 immunotherapy. Altered microbiota in patients with CRC promoted resistance to immunotherapy when transferred to germ-free mice. The study identified that ILC3s interact with T cells via major histocompatibility complex class II (MHCII), which is necessary for supporting type-1 immunity and colonization with beneficial microbiota. Mice with dysregulated ILC3s and altered microbiota exhibited increased tumor progression and resistance to anti-PD-1 therapy. A significant decrease in Th1 cells and T-bet+ CD8 T cells was observed in mice with altered microbiota, leading to impaired type-1 immunity and resistance to immunotherapy. The study indicated that associated inflammation significantly affected gut microbiota composition, leading to immunotherapy resistance [117].

A summary of the role of different microorganisms and associated factors is given in Table 2.

## 7. Conclusions

The burgeoning research on the gut microbiome has unveiled its pivotal role in colorectal cancer (CRC) immune response. This review has systematically highlighted key findings that establish the gut microbiota as a critical modulator of the immune landscape within the CRC microenvironment.

Microbiome composition and CRC immune modulation: The gut microbiome, composed of diverse microbial communities, interacts dynamically with the host’s immune system. Beneficial bacteria such as Bifidobacterium adolescentis and harmful ones like Fusobacterium nucleatum differentially influence immune responses by modulating macrophage polarization, impacting CRC progression and therapy outcomes.

Impact on immunotherapy: Microbial metabolites, particularly short-chain fatty acids (SCFAs), play a significant role in enhancing the efficacy of immunotherapies. SCFAs like butyrate and acetate influence T cell differentiation and function, thereby modulating anti-tumor immunity. Additionally, the gut microbiome’s composition has been linked with the effectiveness of immune checkpoint inhibitors, highlighting the microbiota’s potential as a biomarker for predicting therapeutic responses.

Mechanistic insights: Advances in metagenomic and metabolomic technologies have provided deep insights into the genetic and metabolic pathways through which the microbiome influences host immunity. Key mechanisms include modulation of pattern recognition receptors (PRRs), regulation of tight junction proteins, and production of signaling molecules that collectively maintain gut integrity and modulate immune responses.

Future directions and clinical implications: The integration of microbiome research into clinical practice promises innovative strategies for CRC management. Future research should focus on the following:

Developing microbiome-based biomarkers for early detection and prognosis and immune therapy response of CRC [94,120,121,122].

Exploring microbiome modulation as an adjunct to existing therapies to enhance their efficacy [123,124,125,126,127,128].

Investigating personalized microbiome-targeted interventions considering individual genetic and microbial profiles [129].

In conclusion, the gut microbiome stands as a cornerstone in understanding and improving colorectal cancer prophylactic and immunotherapy. Microbiome interactions with the immune system underscore the necessity for continued research and integration of microbiome science into clinical oncology to pave the way for novel therapeutic avenues and improved patient outcomes.

## Figures and Tables

**Figure 1 cells-13-01437-f001:**
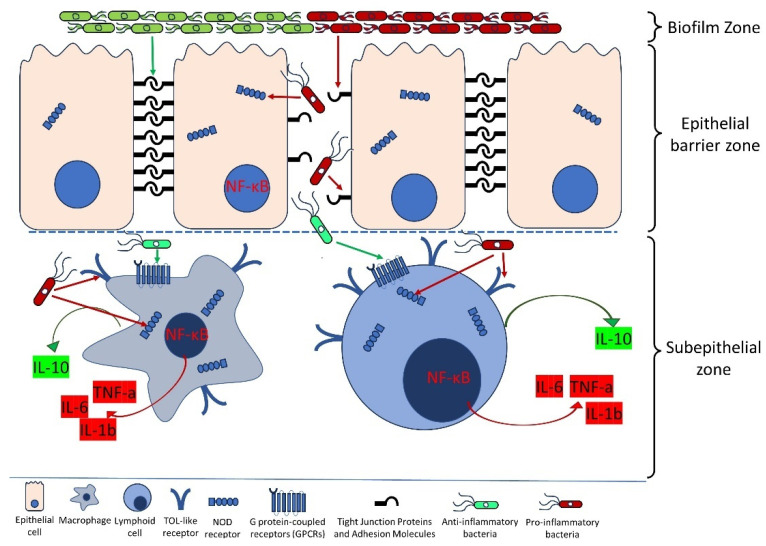
The mechanisms of interaction between the microbiome and innate immunity.

**Figure 2 cells-13-01437-f002:**
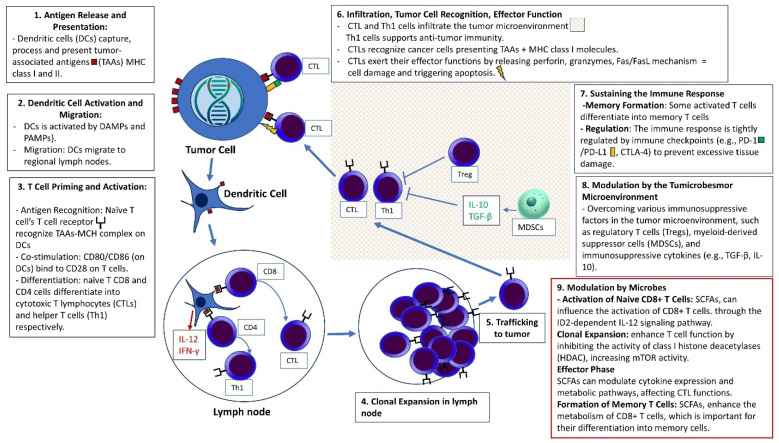
Mechanisms of the anti-tumor immune response against colorectal cancer.

**Table 1 cells-13-01437-t001:** Summary of factors influencing the impact of the gut microbiome on immunity.

Factor	Influence on Gut Microbiome	Impact on Immunity	Key Points	References
Microbial Diversity	High diversity linked with robust immune health	Enhances immune responses, modulates immune cells	Promote SCFA production, crucial for immune regulation and barrier integrity	[17,18,19,20,21,22]
Dietary Influences	Fiber-rich, polyphenol-rich diets support a healthy microbiome; high-fat, high-sugar diets disrupt balance	Fiber and polyphenols enhance anti-inflammatory responses; fats and sugars may increase inflammation	Diet affects microbiome composition, influencing systemic inflammation and immune response	[32,33]
Antibiotics and Medications	Reduce microbial diversity, impact on metabolic functions	Can trigger inflammatory responses or disrupt immune homeostasis	Long-term use of antibiotics linked to negative impacts on microbiome diversity and immune function	[34,35]
Host Genetics	Host genetic factors can determine microbiome composition and diversity	Host genetic factors can affect immune responses mediated by the microbiome	Some genetic markers, such as those for lactase production, correlate with microbiome composition	[36,37,38,39]

**Table 2 cells-13-01437-t002:** Influence of gut microbiota factors on immunotherapy efficacy in colorectal cancer.

Pro-Immunotherapy	Anti-Immunotherapy
Microbes
-*Faecalibacterium prausnitzii* [104]	-*Fusobacterium nucleatum* [103,113]
-Lachnospiraceae family [108]	-*Escherichia coli* [103]
-*Lactobacillus gallinarum* [84]	-Prevotella [103]
-*Coprobacillus cateniformis* [84]	-*Bacteroides* spp. [104,117]
-*Erysipelatoclostridium ramosum* [84]	-Bacillus class [107]
-*Bacteroides thetaiotaomicron* [111]	-Clostridiaceae [116]
-*Bacteroides fragilis* [111]	-Proteobacteria [110]
-*Bacteroides cellulosilyticus* [111]	-*Lactobacillus murinus* [111]
-*Bifidobacterium pseudolongum* [112]	-Lactobacillales order [107]
-*Lactobacillus johnsonii* [112]	
-*Olsenella* species [112]	
-*Blautia* genus [112]	
-*Parabacteroides distasonis* [106]	
-*Alistipes* genus [107]	
-*Eubacterium rectale* [118]	
-*Roseburia* spp. [118]	
-Methanobrevibacter [118]	
-*Akkermansia muciniphila* [119]	
-*Ruminococcus lactaris* [119]	
-*Schizosaccharomyces octosporus* [114]	
-Agathobacter [115]	
Metabolites
-Indole-3-carboxylic acid (ICA) [84]	-Pyrrolidine [107]
-Inosine [112]	
-Pentanoate [97]	
-Short-chain fatty acids (SCFAs) [98,99,108,118]	
-Punicic acid [111]	
-Trimethylamine N-oxide (TMAO) [100]	
Molecular and Cell Mechanisms
-Upregulation of CD80/CD86 on dendritic cells [104]	-Activation of nuclear factor-κB (NF-κB) pathway [103]
-Increased ICOS gene expression in T cells [104]	-Upregulation of cytokines IL-6 and TNF-α [103]
-Accumulation of tumor-specific T cells and memory T cells [105]	-Downregulation of miRNA18a via TLR-4/MYD88 pathway [103]
-High clonality and low diversity in TCR repertoires [103]	-Decreased Th1 cells and T-bet+ CD8 T cells [117]
-Downregulation of PD-L2 expression on dendritic cells [84]	
-Increased systemic CD8α+ dendritic cells and IL-12 levels [101]	
-Enhanced Th1 differentiation and anti-tumor immunity via A2A receptor signaling pathway on T cells [117]	
-Enhanced metabolism and memory potential of CD8+ T cells [98]	
-Increased activity of mTOR pathway [97]	
-Enhanced systemic CD8α+ dendritic cells and IL-12 levels through vancomycin-induced gut microbiome alterations [101]	
-Increased CD3+ and CD8+ tumor-infiltrating lymphocytes (TILs) density [108]	
-Positive modulation by Schizosaccharomyces octosporus of inhibitory receptors PD-1 and CTLA-4 [114]

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
