# Peer review of "Exploring the Role of the Gut Microbiota in Modulating Colorectal Cancer Immunity"

_cells, 2024, doi:10.3390/cells13171437_

Round 1

Reviewer 1 Report

Comments and Suggestions for Authors

The review presents an unbiased summary of the current understanding of the topic.This is an interesting and logical review, which illustrates the interactions between gut microbiota and tumor immune dynamics, with a focus on colorectal cancer.However, When describing the relationship between microorganisms and the colon-associated immune environment, the relationship between the relevant immune factors and specific species or flora can be listed at each node. It would be better if the corresponding treatment measures could be added in relation to the treatment of tumors. In addition, in terms of the structure of the article, Part 2, Foundations of Immuno-Microbial Interplay in Early Life, does not reflect the relationship between tumors and microorganisms at each stage of growth in the subsequent content. Colon cancer is known to not usually occur in adolescents, so this section can be removed. In Section 5, drawing a diagram to illustrate the relationships between Molecular Mechanisms Encoded by Host and Microbial Genes in Gut Microbiome Immune System would be more easy to understanding.

Author Response

Comments 1: When describing the relationship between microorganisms and the colon-associated immune environment, the relationship between the relevant immune factors and specific species or flora can be listed at each node.

Response 1.: Thank you for your comment. I agree with your observation. The names of the microorganism species have been added in accordance with the immune mechanisms with which they are associated according to the literature.

Comments 2: It would be better if the corresponding treatment measures could be added in relation to the treatment of tumors.

Response 2: Thank you for your comment. A new chapter, numbered 6 (The Influence of Gut Microbiota on Immunotherapy Efficacy in Colorectal Cancer), has been added.

Comments 3: In addition, in terms of the structure of the article, Part 2, Foundations of Immuno-Microbial Interplay in Early Life, does not reflect the relationship between tumors and microorganisms at each stage of growth in the subsequent content. Colon cancer is known to not usually occur in adolescents, so this section can be removed.

Response 3: Thank you for your comment. Part 2 (Foundations of Immuno-Microbial Interplay in Early Life) has been removed from the manuscript.

Comments 4: In Section 5, drawing a diagram to illustrate the relationships between Molecular Mechanisms Encoded by Host and Microbial Genes in Gut Microbiome Immune System would be more easy to understanding.

Response 4: Thank you for your comment. A corresponding illustration has been added to Chapter 4 (formerly Chapter 5, as the chapter numbering changed after the removal of the second chapter).

Reviewer 2 Report

Comments and Suggestions for Authors

Sharkhpazyan et al. provided a detailed and comprehensive review on the role of gut microbiota in modulating colorectal cancer immunity. The manuscript describes recent findings in the field and discusses future questions that need to be addressed. The work is logical, well-constructed, and interesting for readers.

Note: There are minor errors throughout the manuscript that need to be corrected.

Comments on the Quality of English Language

Small tapes that need to be repair

Author Response

Comments: There are minor errors throughout the manuscript that need to be corrected.
Small tapes that need to be repair

Response: Thank you for your comment and for evaluating our work. The manuscript has been improved with new information on the influence of microbiota on immunotherapy, and a new illustration has been added.

Reviewer 3 Report

Comments and Suggestions for Authors

The manuscript entitled “Exploring the Role of the Gut Microbiota in Modulating Colorectal Cancer Immunity” aims to span foundational concepts of immuno-microbial interplay, factors influencing microbiome composition, and evidence linking gut microbiota to cancer immunotherapy outcomes. It should be thoroughly revised before being published because some parts of the manuscript are not consistent with the subject of the review. For example, part “2. Foundations of Immuno-Microbial Interplay in Early Life” is not quite relevant to the subject. Colorectal cancer is commonly found in adults while not in children.

Instead, the authors are encouraged to supplement detailed information regarding the mediation effects of gut microbiome on cancer immunotherapy, with particular focus on colorectal cancer. This should be considered as one of the pivotal parts of this review. Readers would expect to find real-world data and to know the implications from previous work in the paper.

Finally, fig 1 is too simple to illustrate the interactions between gut microbiome and colorectal cancer immunity. Since this is the only figure in the paper, it should focus on the specific characteristics of colorectal cancer and the detailed pathways involving in the interactions. Therefore, the figure should avoid using unspecific expressions like “myeloid-derived suppressor cell” and should include networks if possible.

Comments on the Quality of English Language

Readable.

Author Response

Comments 1: For example, part “2. Foundations of Immuno-Microbial Interplay in Early Life” is not quite relevant to the subject. Colorectal cancer is commonly found in adults while not in children.

Response 1: Thank you for your comment. Part 2 (Foundations of Immuno-Microbial Interplay in Early Life) has been removed from the manuscript.

Comments 2: The authors are encouraged to supplement detailed information regarding the mediation effects of gut microbiome on cancer immunotherapy, with particular focus on colorectal cancer. This should be considered as one of the pivotal parts of this review. Readers would expect to find real-world data and to know the implications from previous work in the paper.

Response2: Thank you for your comment. A new chapter, numbered 6 (The Influence of Gut Microbiota on Immunotherapy Efficacy in Colorectal Cancer), has been added.

Comments 3: Finally, fig 1 is too simple to illustrate the interactions between gut microbiome and colorectal cancer immunity. Since this is the only figure in the paper, it should focus on the specific characteristics of colorectal cancer and the detailed pathways involving in the interactions. Therefore, the figure should avoid using unspecific expressions like “myeloid-derived suppressor cell” and should include networks if possible.

Response3: Thank you for your comment. Another figure has been added to the manuscript, and the figure you commented on is now numbered as Figure 2 after the revisions. Following your recommendations, the figure has been modified to include more details, illustrating the nature of the immune response in colorectal cancer and the factors influencing it, including microbial effects.